# Chemistry and Bioinformatics Considerations in Using Next-Generation Sequencing Technologies to Inferring HIV Proviral DNA Genome-Intactness

**DOI:** 10.3390/v13091874

**Published:** 2021-09-19

**Authors:** Guinevere Q. Lee

**Affiliations:** Division of Infectious Diseases, Department of Medicine, Weill Cornell Medicine, New York, NY 10021, USA; gul4001@med.cornell.edu

**Keywords:** HIV genomes, HIV persistence, deep sequencing

## Abstract

HIV persists via integration of the viral DNA into the human genome. The HIV DNA pool within an infected individual is a complex population that comprises both intact and defective viral genomes, each with a distinct integration site, in addition to a unique repertoire of viral quasi-species. Obtaining an accurate profile of the viral DNA pool is critical to understanding viral persistence and resolving interhost differences. Recent advances in next-generation deep sequencing (NGS) technologies have enabled the development of two sequencing assays to capture viral near-full- genome sequences at single molecule resolution (FLIP-seq) or to co-capture full-length viral genome sequences in conjunction with its associated viral integration site (MIP-seq). This commentary aims to provide an overview on both FLIP-seq and MIP-seq, discuss their strengths and limitations, and outline specific chemistry and bioinformatics concerns when using these assays to study HIV persistence.

## 1. Introduction

HIV-1 infection leads to lifelong viral persistence. Upon infection, the viral RNA genome is reverse transcribed the into viral cDNA, which is followed by an irreversible integration into the human genome [1]. This results in the establishment of a viral DNA reservoir that fuels subsequent viral replication cycles when treatment is stopped [2,3,4,5]. Antiretroviral treatment is effective in suppressing ongoing viral replications but does not eliminate integrated HIV genomes. As such, HIV-infected individuals rely on lifelong treatment to suppress ongoing viral replications. In the absence of treatment, viral replication resumes, new CD4^+^ cells are infected, and infected individuals progress to develop acquired immunodeficiency syndrome (AIDS) [6]. 

The viral DNA reservoir that sustains HIV persistence is extremely stable in size and has been estimated to have a half-life of approximately 44 months in individuals receiving long term antiretroviral treatment [5], suggesting the general inability of the immune system to naturally clear the viral reservoir despite suppression of active viral replication. Recent studies have revealed that viral persistence is maintained, at least partially, by clonal expansion of infected cells [7,8,9,10,11,12,13,14,15], which is driven by mechanisms such as antigenic stimulation [9,16], homeostatic proliferation [15,17], and proliferation associated with host genomic locations of viral integration [10,18]. The relative abundances of these clonally expanded infected cells wax and wane over time [15]. In other words, the viral DNA pool within an individual is not a static population and has a relatively slow half-life. 

The viral DNA pool is also genetically diverse: even during hyperacute heterosexually-transmitted HIV infection, when a single founder virus is presumed [19], each HIV-DNA genome contains at least one single-base nucleotide substitution mutation [20], presumably attributable to the high error rate of the HIV reverse transcriptase. Genetically similar but non-identical viral sequences within an infected individual are called “viral quasi-species”. In addition, a close examination of the genotypic compositions of HIV DNA in chronically infected individuals revealed that over 90% of the viral DNA genomes are heavily truncated, have deleterious insertions/deletions, have been excessively hypermutated, and/or have single-base substitution mutations that would yield premature stop codons in essential viral genes [21,22]. Viral genomes that contained such decapacitating alterations are incapable of fueling virologic rebound in the absence of treatment and have been termed “defective HIV-DNA genomes” as opposed to “genome-intact HIV-DNA genomes,” which lack obvious defects. Furthermore, each HIV-DNA genome is also integrated into distinct locations within the host chromosome, creating another factor that contributes to reservoir population diversity. Recent studies have suggested that integration sites of genome-intact HIV proviruses into transcriptionally less-active human chromosomal regions may be associated with a decreased likelihood of viral transcription activation [23,24]. 

In summary, the HIV reservoir population structure within a single infected individual is complex, changes over time, and contains viral genomes that are either intact or defective, while each viral genome is associated with unique viral integration sites that may impact their likelihoods of transcription activation. To study viral persistence and its longitudinal dynamics and to identify future targets for HIV cure research, it is therefore crucial to accurately characterize “genome-intact” HIV-DNA genomes. In this commentary, the author will discuss technical considerations and limitations of two assays, FLIP-seq and MIP-seq, both of which are single-copy, next-generation deep sequencing techniques for the study of HIV DNA genomes and reservoirs. 

## 2. Traditional Assays and the Subsequent Development of FLIP-Seq and MIP-Seq

Traditional techniques for the study of HIV reservoirs include cell-culture-based quantitative viral outgrowth assay (qVOA), quantitative short-amplicon PCR (qPCR), and droplet digital PCR (ddPCR). Each of these assays has merits and limitations: qVOA relies on the stimulation of infected cells ex vivo and detection of viral RNA production in culture supernatant and measures true replication-competent and genome-intact proviruses but is labor-intensive, and it has been shown that a single round of activation is insufficient to reactivate all replication-competent genome-intact proviruses [13]. Both qPCR- and ddPCR-based total HIV-DNA reservoir sizes quantification approaches amplify and quantify short regions of the viral DNA genome and are relatively inexpensive and scalable but do not distinguish between intact versus defective HIV-DNA genomes [25,26]. 

In 2013, using HIV near-full-genome Sanger sequencing, a paradigm-shifting study by Ho et al. [21] showed that the vast majority of viral DNA in infected individuals are genome-defective, prompting the HIV persistence and cure research community to shift focus onto identification of cells infected with genome-intact proviruses. In 2017, two groups independently developed and published next-generation deep sequencing versions (as opposed to Sanger sequencing [21]) for near-full-genome HIV-DNA sequencing [7,8], nowadays known as FLIPS or FLIP-seq. Since then, multiple research groups also have, based on these existing proviral full-genome sequence data, developed new qPCR or ddPCR methods, such as the ddPCR-based Intact Proviral DNA Assay (IPDA) [27], a hybrid qPCR/sequencing method Q4PCR [28], and a ddPCR method by Levy et al. [29], all of which use multiple probes and multiplexing to infer and quantify intact versus defective proviral genome status. This article will focus on evaluating single genome amplification and sequencing methods and will use the term FLIP-seq to refer to the assay as published by [7], but the biochemical and technical considerations discussed below may be applied to any single genome amplification and sequencing assays. 

Briefly, similar to the 2013 Sanger sequencing method [21], FLIP-seq (Figure 1a) starts with a DNA extraction of an infected cell population (further discussed in Section 2), followed by a rough quantification of total HIV-DNA copies by either qPCR/ddPCR or serial dilution to estimate the copy numbers of total HIV-DNA genomes concentration per extraction volume within the sample. Using this concentration estimate, limiting dilution of the DNA extract is performed by diluting the extract to one HIV-DNA template-positive per three PCR reactions according to Poisson distribution (Section 3). Then, using HIV-specific primers validated for subtype B [7], C [20], and D [30] HIV-1, each reaction is subjected to PCR (Section 4) to amplify near-full-genome HIV DNA. Since each PCR-positive reaction contains the amplification products originated from approximately a single HIV-DNA molecule, this set up is termed “single genome amplification” (SGA). Resulting amplicons are each subjected to next-generation sequencing library preparation and tagging using unique molecular indexes, then pooled and deep sequenced (Section 4). Resulting short reads are demultiplexed, de-novo assembled, and subjected to bioinformatics inferences on genome-intactness (Section 5). 

FLIP-seq yields high-resolution HIV-DNA genome sequences; however, to study the integration site of genome-intact proviruses, another technological breakthrough was needed: traditional techniques to examine HIV integration sites involves Sanger or deep sequencing of the viral-host junctions [31]. This approach is scalable, but targeting a short genomic region around the viral-host junction did not allow for the discrimination of integration sites associated with genome-intact versus defect HIV DNA. In 2019, two groups independently published deep sequencing methods to co-sequence full-genome HIV DNA and viral integration sites. These assays were named MIP-seq [32] and MDA-SGS (Multiple Displacement Amplification Single Genome Sequencing) [33], respectively. This article specifically focuses on evaluating biochemical and technical considerations of MIP-seq [32], but the considerations discussed below may be applied to both assays. 

Similar to FLIP-seq, MIP-seq (Figure 1b) starts with DNA extraction of an infected cell population (Section 2) and limiting dilution (Section 3). Each single viral genome dilution aliquot is then subjected to multiple displacement amplification (MDA) in order to unbiasedly amplify all DNA genetic materials within the aliquot. This reaction is then split into two portions: one of which is subjected to a five-overlapping-amplicon HIV genome PCR amplification (Section 4), and the other portion is subjected to viral-human DNA junction amplification. All resulting amplicons are deep sequenced (Section 4), followed by bioinformatic inferences of viral genome intactness and the identification of viral integration sites coordinates within the human genome (Section 5).

## 3. DNA Extraction

One of main purposes of both FILP-seq and MIP-seq is to capture genome-intact HIV DNA, which is approximately 10,000 base pairs in length [34]. It is therefore crucial that the chosen extraction method does not introduce extensive shearing of DNA templates to below the target capture length. As different extraction methods introduce different DNA shearing profiles [35,36], the choice of extraction method will impact assay sensitivity in terms of full-length viral genome recovery. Another factor that impacts recovery is the extraction mechanism: column-based commercial extraction kits are known to have lower overall DNA recovery compared to magnetic bead-based methods [37]. Other factors, such as incubation method, time, and temperature, also impact percentage shearing and recovery [38]. 

To monitor DNA shearing and assay recovery and to ensure assay reproducibility, it is therefore necessary to implement quality control protocols. Two methods will be discussed below: the first is using Agilent Bioanalyzer systems or similar technologies. For example, the Agilent 2200 TapeStation is a chip-based capillary electrophoresis system that will analyze the DNA fragment-size distribution in a given sample [39]. Nucleic acid extractions prepared for FLIP-seq and MIP-seq processing could be analyzed via similar platforms to ensure the presence of fragments around 10,000 base pairs to ensure maximal recovery of genome-intact proviral genomes. 

The second method is complementary and involves the use of a positive control with assay-specific primers. The positive control can be any known HIV-DNA material that has known clonal full-length viral genomes. One example is a cell line called 8E5/LAV (NIH AIDS Reagent Program Catalog #95 [40]), which has roughly a single copy of integrated full-length HIV genome per cell. After nucleic acid extraction, the sample is split into two aliquots: one is subjected to limiting dilution and SGA short-amplicon HIV-specific PCR amplification (e.g., *pol*), whereas the other aliquot will be subjected to the same limiting dilution factor identical to the short-amplicon reactions but will be amplified for near-full-length viral genomes in the case of FLIP-seq or subjected to the five amplicon PCR approach in the case of MIP-seq. The ratio between the recovery in the full-genome amplification approaches relative to the short amplification approach would reveal the assay sensitivity against the shorter amplification region. Note, this quality control method measures comprehensive assay sensitivity that includes both DNA shearing and PCR DNA polymerase efficiency, which will be discussed in Section 4 below.

Finally, in light of variabilities in recovery and extent of template shearing depending on extraction methods, it is important to restrict any FLIP-seq- and/or MIP-seq-based quantitative comparisons across samples and/or cohorts to samples that were processed using identical DNA-extraction methods. It is also important to note that FLIP-seq and MIP-seq are theoretically only semi-quantitative at best, a concept which will be further explored in Section 4.

## 4. Poisson Distribution and Limiting Dilution

Both FLIP-seq and MIP-seq involve limiting dilutions of the nucleic acid extract to achieve single-genome amplification (SGA) by PCR. There are at least three main reasons why SGA should be strictly enforced: the first and perhaps the most important reason is PCR efficiency [41,42]. If a short, truncated, and defective HIV-DNA genome is present in the same PCR reaction well together with a long, intact HIV genome template, amplification efficiency will be higher for the short relative to the longer genome, resulting in a bias of short genome detection. The second reason is to reduce the likelihood of inter-template recombination, which is a well-described PCR phenomenon [43,44,45]. The third reason is to resolve viral quasi-species. HIV is genetically diverse due to an error-prone reverse transcriptase, which introduces approximately one error into the viral genome at every viral RNA to DNA conversion step [46]. These mutations accumulate over the course of active viral replication and create a genetically diverse within-host viral quasi-species population that allows for Darwinian selection for drug resistance [47] and/or immune escape [48] variants. Given that every HIV-DNA template is potentially genetically different (with the exception of clonally expanded proviral populations), SGA ensures that even single-base differences would be clearly resolved. Note, resolution also depends on PCR fidelity (further discussed in Section 4). 

The rule of thumb in setting up limiting dilutions for both FLIP-seq and MIP-seq is to achieve one PCR-positive reaction in every three reactions, or a “1 in 3” setup, or an SGA ratio of 0.3, to yield a Poisson probability of 85.7% that a given PCR-positive well has originated from a single HIV-DNA molecule. Figure 2 illustrates the theoretical relationship between varying SGA ratios and the probability of single-molecule amplification. Referring to Figure 2, shifting the SGA ratio to a “1 in 2” setup would result in a 77.1%, whereas a “1 in 1” setup would result in a 58.2% Poisson probability of having one template of origin per positive PCR reaction. In contrast, a “1 in 100” setup would result in a 99.5% probability of single-genome amplification. Given these probability values, a researcher setting up SGA reactions should strike a balance between reagent cost and data quality, as increasing the number of amplicon-negative wells dramatically increases PCR reagent costs. The SGA ratio of 0.3 or a Poisson probability of 85.7% is an arbitrary value generally accepted by the research community [49]. 

Since a “1 in 3” setup only yields 85.7% probability of single-template amplification, there is a 14.3% probability that a given PCR-positive reaction under this setup could have originated from multiple HIV DNA molecules. Deep sequencing of each PCR-positive reaction allows for post-hoc bioinformatic evaluation of whether there are multiple HIV-DNA species present in the reaction: Applicable to both FLIP-seq and MIP-seq, the presence of base-pair mixtures per genome position at frequencies above the expected sequencing error rate serves as an indicator of the presence of multiple DNA templates. In the case of MIP-seq, presence of multiple HIV integration sites by deep sequencing also marks the potential presence of multiple input templates. Depending on the research question, these multiple-template PCR positive reactions could be removed from the final data analyses to achieve maximal data quality. 

Note also that the distribution shown in Figure 2 assumes 100% assay sensitivity; in other words, the probability of 85.7% for single-template amplification is achieved only if every single template input, both long and short, was successfully amplified at 100% PCR efficiency. As discussed above, PCR efficiency varies according to template lengths. Therefore, even PCR-negative wells could have contained a HIV template that was not successfully amplified. This implies that the traditional dilution-factor calculation approach by visually counting the number of PCR-positive reactions by gel electrophoresis after near-full viral genome amplification and then selecting a dilution factor that yields 1 in 3 visually detectable amplicons could possibly lead to under dilution, mainly due to the lower PCR efficiency against longer input HIV-DNA templates. A potential solution to this issue is to calculate the dilution factor for limiting dilution using the concentration of total HIV-DNA genomes derived from a short target region amplification (e.g., via ddPCR amplification of a short, conserved region in the HIV genome [7]) to achieve a higher PCR efficiency relative to full-genome long template amplification. 

Given that each HIV-infected study participant has a distinct profile of HIV-DNA genome lengths [7,8,20,21,22,50] and given that PCR efficiency is not identical across varying template lengths and that SGA ratios are at best an estimate, plus the fact that PCR-DNA polymerase activity decreases over storage time, both FLIP-seq and MIP-seq should be considered only semi-quantitative with a bias towards detection of shorter viral genomes. In addition, due to the low frequency of productively infected CD4^+^ cells in long-term, antiretroviral-treated, HIV-infected donor samples, typically estimated to be approximately one per million CD4^+^ [5,51], when compounded with imperfect PCR efficiencies, approximately 10 million CD4^+^ cells are typically required in order to detect at least one intact genome per donor. Therefore, despite their high resolution, sample availability can be a major challenge when using FLIP-seq and MIP-seq for genome-intact virus quantification.

## 5. PCR Fidelity and Sequencing Errors

DNA polymerases, such as Taq, used in PCR reactions and sequencing library preparations can introduce errors in amplification products [52]. PCR fidelity refers to the accuracy of bases incorporated [42]. A high-fidelity DNA polymerase results in a low error amplification profile. Errors can also be introduced by the sequencing process itself: for example, Illumina sequencing is reported to have a baseline sequencing error rate of approximately 0.2% [53]. As discussed in Section 3, each HIV-DNA genome can potentially harbor at least one single-base nucleotide mutation due to the error prone viral reverse transcriptase [46]. Based on this observation, identical HIV-DNA genome sequences obtained from SGA reactions and FLIP-seq are often used as markers for the clonal expansion of infected cells [7,8,50]. The validity of using near-full-genome FLIP-seq sequence-identity to mark clonal expansion has been further supported by later observations from MIP-seq, showing that 100% identical viral sequences also have identical viral-host integration junctions [32]. This implies that PCR and sequencing errors should be strictly monitored for any viral-sequence-based clonal expansion analyses that are not supported by viral integration site data. In addition, PCR and/or sequencing errors can also introduce artificial stop codons into a proviral genome, leading to false classification of genome defectiveness. As such, it is important to optimize both FLIP-seq and MIP-seq to yield the most accurate viral genome sequence data possible.

The first optimization step is to select a DNA polymerase with high fidelity for viral genome amplification: FLIP-seq, as published in [7], uses a third-generation Invitrogen Taq polymerase (catalog number 11304102) at 6X fidelity relative to unmodified regular Platinum Taq [54]. Since no PCR amplification is completely error-free, but errors introduced are relatively random in terms of kinds (base substitutions, deletions, and insertions) and locations [55], it is possible to bioinformatically correct for errors given a deep enough sequencing depth via the generation of consensus sequences (Figure 3). The median sequencing depth (a.k.a. coverage) across the HIV genome for previously published FLIP-seq [56] and MIP-seq [32] data was at approximately 2000 Illumina small reads (150 bp) per base position.

To bioinformatically measure and/or correct for errors, a consensus viral genome sequence is first generated from deep sequencing reads derived from an SGA reaction, then the distribution and prevalence of non-consensus base pairs across the viral genome is calculated. Note that under the same principal, this consensus-based correction method not only corrects for PCR errors but also serves to correct errors introduced by MDA (in the case of MIP-seq), various sequencing library preparation protocols that are PCR-dependent, as well as the errors introduced during the process of sequencing itself. This error-detection step is an integrated part of the bioinformatics pipeline HIVSeqinR [20] developed for HIV proviral genome-intactness inferences. Another cross-validation for undetectable PCR/sequencing error post-bioinformatics correction, as mentioned above, is through the identification of 100% genetically identical viral DNA genomes in addition to identical MIP-seq-derived integration site coordinates [32]. 

Note that the above discussion applies only to viral genome sequencing. PCR fidelity is less critical in integration site sequencing and mapping. This is because the viral-host junction sequence data are only used for mapping to the human reference genome for the identification of integration site as opposed to quasi-species differentiation. In one of the MIP-seq algorithms as published in [32], during this step, query fragment lengths in blocks of 20 nucleotides [57] would be evaluated by the mapping algorithm, making the results less susceptible to single-base substitutions, insertion, and deletion errors associated with PCR enzyme fidelity and sequencing errors. 

## 6. Bioinformatics Considerations for Genome-Intactness Inferences

Viral genomes captured by FLIP-seq and MIP-seq are typically subjected to bioinformatics evaluation for genome-intactness. The term “genome intactness” is loosely defined as the lack of any decapacitating mutations that would render a viral genome non-replication competent. However, the exact definitions/criteria vary between research groups and publications [7,8,20,21,27,33,50]. A few common categories of “genome defects” will be discussed below; criteria used in the automated genome-intactness calling computational pipeline HIVSeqinR [20] will be given as examples. A stable release (version 2.7.1 as of date of manuscript preparation) is available in GitHub at https://github.com/guineverelee/HIVSeqinR (accessed on 7 September 2021). Another software for HIV-DNA genome-intact inferences, HIVIntact [58], is also publicly available for download and differs from HIVSeqinR in terms of logical order for intact determination as well as specific bioinformatic definitions of “genome defects”. Regardless of the software used, a bioinformatics inference strategy should aim to optimize sensitivity and specificity for the purpose of a specific research question. For example, the software HIVSeqinR [20] was designed to maximize specificity in genome intactness, calling to predict replication competency. A classification algorithm should also be reproducible; in other words, ideally the same inference strategy should be applied to all viral genome sequences in a dataset. 

### 6.1. Large Deletions

A viral genome may be heavily truncated, rendering it non-replication competent. These are genomes that contain “large deletion(s)”. However, “large” is a relative term. In addition, deletion(s) and/or truncation(s) that occurs within an essential gene may directly impacts replication competency. In HIVSeqinR, any near-full-length HIV amplicons less than 8000 bp are automatically categorized as having “large deletions”. In other words, assuming an amplicon spanning HXB2 coordinates 638–9632 [7], any genomes with deletion(s) more than approximately 995 bp relatively to the 8995 bp expected length will be classified as a genome with “large deletion(s)” regardless of the location and frequencies of truncation. Note that this strategy is designed to achieve automation, reproducibility, and to maximize specificity against the detection of a replication-competent virus when used in combination with the other defectiveness categories. 

### 6.2. Internal Inversions

A portion of the viral DNA genome may contain an inversion, rendering it non-replication competent. In HIVSeqinR, inversions are detected by mapping query sequences at an initial block/window size of 11 bp [59]. Adjusting this length can impact the sensitivity of internal inversion detection. 

### 6.3. Hypermutation

Guanosine to adenosine (G-to-A) hypermutations are introduced into viral genomes during the reverse transcription step by a family of host-defense proteins called APOBEC, leading to the occurrences of premature stop codons throughout the genome [60]. A web tool called Hypermut [61] is available in the Los Alamos HIV Sequence Database website to screen whether a given query genome contains APOBEC-associated footprints. This algorithm is reference-sequence dependent: briefly, it counts the occurrences of where Gs are expected based upon the reference genome that the user uploaded. For the most accurate prediction, a donor-matched reference sequence that has been shown to be replication competent experimentally should be used, but this sequence is often not available. In HIVSeqinR’s adaptation of Hypermut [20], HXB2 is used as the universal reference sequence to provide a baseline screen for genomes that have obvious APOBEC footprints; all other genomes that contain a large amount of premature stop codons would be identified as having “premature stop codons” at a later stage in the HIVSeqinR algorithm and will not be classified as intact. In other words, HIVSeqinR compromises on sensitivity for true APOBEC-associated hypermutated genomes in return for automation with a focus on maximizing overall specificity for genome-intact inferences. If the purpose of one’s research is, for example, not to identify intact genomes but to study the impact of APOBEC protein family on HIV DNA reservoirs, then it becomes important to fine-tune this hypermutation inference process as an independent, non-automated step using the most appropriate reference genome available. 

### 6.4. Premature Stop Codons

A viral genome may contain single-base substitution mutation(s) and/or out-of-frame insertion/deletion(s), rendering the genome non-replication competent. Three main considerations should be given when evaluating a specific genome for this category. First, location of a given premature stop codon matters: the HIV genome codes for nine genes (*gag*, *pol*, *vif*, *vpr*, *vpu*, *tat*, *rev*, *env,* and *nef*), while only *gag*, *pol*, and *env* are traditionally considered essential genes [34]. There are known examples of HIV genomes with premature stop codons in *tat* and *nef* that are able to establish infections both *in vitro* (for example, *tat* [62] and Table 1; *nef* [63,64]) and *in vivo* (for example, *tat* [65]; *nef* [66,67]) despite reduced function/replication capacity [63,65,66]. In HIVSeqinR [20], a viral genome is labelled to contain “premature stop codon(s)” only if the stop codon occurs in any one of the essential genes *gag, pol,* and/or *env*. Second, “premature” is a relative term: for instance, a premature stop codon that results in the loss of 50% of the expected amino acid length will have a more decapacitating effect relative to a stop codon that results in the loss of 5% amino acid length. In HIVSeqinR [20], a genome will be labelled to contain “premature stop codon(s)” if the stop codon results in an amino acid length of less than 95% relative to HXB2/JR-CSF/NL4-3 in any of the essential genes *gag, pol,* and/or *env* (Table 1, expected values). This 95% cutoff value maximizes specificity for genome-intactness inferences. However, these definitions are not absolute and should not be considered 100% predictive of replication competency and should be adapted and evaluated for each scientific question being asked. Finally, it is important to ensure stop codons have not been introduced due to PCR and/or sequencing errors. It is therefore important to perform quality control measures as outlined in Section 4: SGA and sequencing of a clonal population should lead to identical consensus sequences. 

### 6.5. 5′ or Psi (ψ) Defects

The 5′ beginning of the HIV genome contains a packaging signal also called ψ (HXB2 coordinates 681–789) [34]. This region has been shown to be essential for viral genome dimerization, nucleocapsid (NC) protein binding, and subsequent viral RNA packaging into viral particles [68]. ψ is non-coding, consists of four stem loops (SL1-4), and depends on the RNA 3D secondary structure to achieve its functions [69,70,71]. There are currently no algorithms available to accurately predict the RNA 3D structure of a given ψ DNA sequence and to distinguish between functional versus defective ψ. For this reason, HIVSeqinR, for example, imposes a loose definition for ψ defects: given that NL4-3 is replication competent [72] and thus has a functional ψ, 5′ defect in HIVSeqinR has been defined as any viral genomes with a ≥15 bp insertion and/or deletion in that region relative to NL4-3 ψ, which is identical to HXB2 ψ, which are both 112 base pairs in length. Again, this definition aims to achieve maximal specificities for genome-intactness predictions based on our knowledge of a replication competent viral strain. 

### 6.6. One Verdict per Genome

First, it is important to note that this above list of potential defect-genome categories is not exhaustive: other definitions can also be considered, such as the presence/absence of splice donor 1 (D1) site [50]. Second, it is possible that one genome contains multiple classes of defects: for example, it is not uncommon to observe genomes with large deletions that are also hypermutated [7]. In HIVSeqinR, for reproducibility and downstream statistical purposes, after obtaining a TRUE/FALSE classification of each of the above defective categories described, each viral genome is then given a single verdict in the order of large deletions, internal inversions, hypermutations, premature stop codons, and 5′ or psi (ψ) defects. Any genomes without any of the above-mentioned defects would be classified as “genome-intact” by HIVSeqinR. Multiple verdict calling is supported by HIVSeqinR by reviewing the raw per-category TRUE/FALSE output. Note that since the purpose of the HIVSeqinR software was to identify genome-intact proviruses, which is a category derived by elimination, therefore, by definition, it is the only classification category that does not support multiple verdicts.

### 6.7. Functional Validation

Any bioinformatics-inference algorithms offer only predictions and should be functionally validated. In the case of proviral genome intactness, the corresponding functional data can be one or both of (i) SGA sequence data of full-genome plasma virus assuming that plasma derived sequences are replication competent and/or (ii) SGA sequence data from assays that measure replication competence, such as qVOA. For example, HIVSeqinR was functionally validated to be 100% sensitive in predicting genome-intactness, qVOA-derived outgrowth viral sequences [7]. Finally, it is important to understand that replication competence is a spectrum: mutations in different parts of the viral genome may increase/decrease the replication fitness of the virus to different degrees. 

In summary, this section highlights that the term “genome intactness” is a strictly bioinformatic definition for the lack of specific defects in a given HIV DNA genome. Researchers should adapt a definition of genome intactness that best suits their specific research question.

## 7. Conclusions

Both FLIP-seq and MIP-seq are deep sequencing assays designed to distinguish between intact versus defective HIV proviral DNA genomes. FLIP-seq and similar technologies have been applied to cross-sectionally examine viral reservoir landscapes in various CD4^+^ T-cell subsets [7,8] and to longitudinally examine the evolution of the viral DNA genome populations over time [20,50]. MIP-seq has been applied to compare viral integration sites of intact versus defective genomes [32], reveal unique patterns of genome-intact viral integration sites in HIV elite controllers [24], and has been further developed by another group of researchers to include co-capturing of T-cell receptor sequences for antigen specificity inferences of the infected cells [73]. Application of these sequencing technologies to various cohorts have resulted in a rich collection of HIV-DNA genome sequences archived in public repositories, such as the HIV Proviral Sequence Databases [73] and the Los Alamos HIV Sequence Database [74], which are used in part to guide the design of relatively low-cost ddPCR-based assays, such as IPDA [27] and a multiplex assay by Levy et al. [29] for the quantification of intact versus defective HIV-DNA genomes. In summary, this commentary highlights that deep sequencing like FLIP-seq and MIP-seq offers advantages, such as high-resolution data quality enabling post-hoc quality control for true single-genome amplification; but in order to take full advantage of these technologies, one has to be mindful to take necessary quality control steps to monitor data quality. The list of chemistry and bioinformatics considerations discussed in this commentary is by no means exhaustive and should be re-evaluated with a given scientific question a researcher sets out to address.

## Figures and Tables

**Figure 1 viruses-13-01874-f001:**
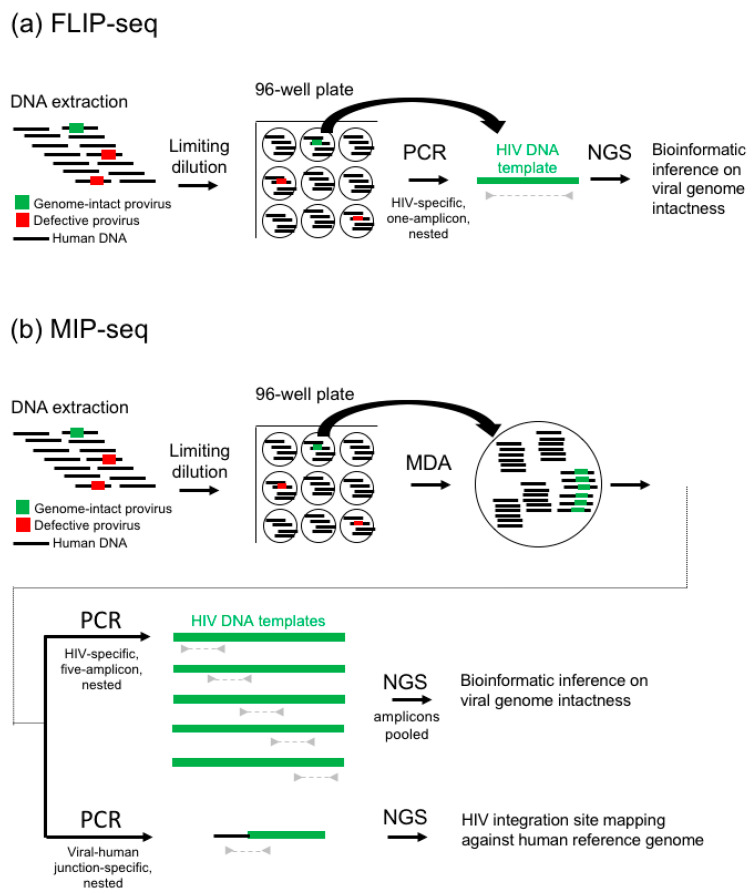
FLIP-seq and MIP-seq workflow. (**a**) FLIP-seq begins with DNA extraction (further discussed in Section 2), followed by limiting dilution to a single HIV-DNA template per subsequent PCR reaction (Section 3), near-full-genome single-amplicon HIV-DNA PCR amplification (Section 4), and finally next-generation deep sequencing (NGS) and bioinformatic inference on viral-genome intactness (Section 5). (**b**) Similar to FLIP-seq, MIP-seq begins with DNA extraction (Section 2), followed by limiting dilution to achieve single HIV-DNA template per subsequent reaction (Section 3), then multiple displacement amplification (MDA) by random primers. Resulting reaction is split for near-full-genome five-overlapping-amplicon HIV-DNA PCR amplification and viral-human junction amplification (Section 4), then subjected to NGS and bioinformatics inference on viral-genome intactness and mapping of viral-human DNA junctions against the human reference genome (Section 5). Note that MIP-seq would not yield full-length sequences of defective viral genomes that do not contain any of the primer binding sites targeted by the 20 primers used in the five-amplicon nested-PCR approach and was designed specifically to capture near-full-length HIV DNA that are approximately >8000 base pairs in length. A single-amplicon, near-full-genome PCR approach was not used in MIP-seq because it was markedly less sensitive due to the average amplification product lengths at the MDA step.

**Figure 2 viruses-13-01874-f002:**
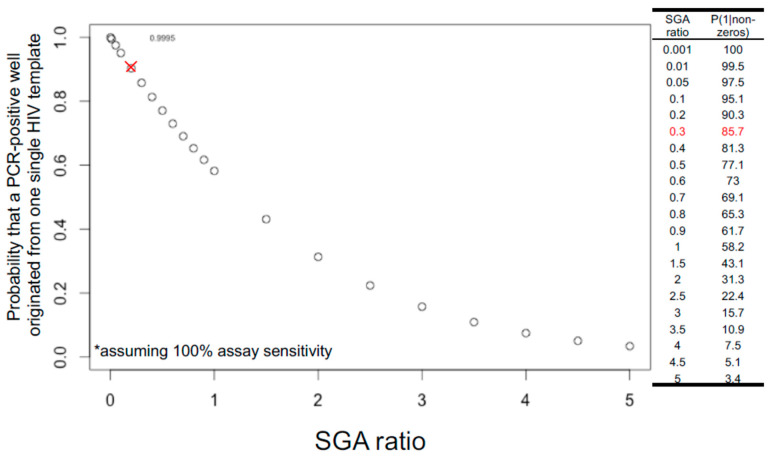
Probability that a PCR-positive reaction originated from one single HIV template follows Poisson probability distribution and decreases as SGA ratio increases. SGA ratio (x-axis) is defined as the number of expected positive reactions divided by the total reactions. For example, if ddPCR short amplicon estimation shows that there are 3 HIV-DNA copies per microliter within a nucleic acid extract, to achieve a limiting dilution at 1:3 (0.3) SGA ratio, one microliter of this extract would be distributed into 9 PCR reactions equally, yielding a probability of 85.7% (y-axis) that a given positive well is derived from one single HIV template (red cross in graph and red highlight in table). This calculation assumes 100% assay sensitivity in the amplification and detection of all input templates (asterisk).

**Figure 3 viruses-13-01874-f003:**
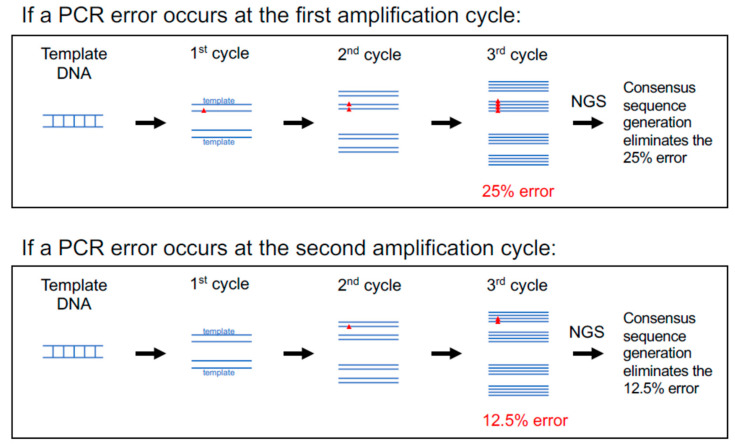
PCR errors are corrected via consensus sequence generation. Errors introduced during the PCR amplification step (red triangles) mainly involve single-base substitution errors at random locations. An error that occurs at an earlier PCR amplification cycle will carry over into a larger proportion within the final amplicon pool (25% if first cycle) relative to an error that occurs at a later amplification cycle (12.5% if second cycle). This figure illustrates that per error introduced, the maximum frequency of representation is 25% within the final amplicon pool, which can be corrected via consensus sequence generation of the deep sequence data.

**Table 1 viruses-13-01874-t001:** Amino acid lengths of all HIV gene products and the lengths of the non-coding packaging signal in five commonly used lab/reference strains are summarized below.

	By Lengths						By Percentages Relative to Expected Values				
Strains	ACH-2	8E5/LAV	HXB2	JR-CSF	NL4-3	(HIVSeqinR Expected Value Settings)	ACH-2	8E5/LAV	HXB2	JR-CSF	NL4-3
NIH HIV Reagent Program ID	ARP-349 **	ARP-95 **	NA ***	ARP-394	ARP-114 **		ARP-349	ARP-95	NA	ARP-394	ARP-114
Replication competence	Yes	No	Weak	Yes	Yes		Yes	No	Weak	Yes	Yes
**Non-coding** **(unit, nucleotide length)**											
Psi length, HXB2 681-789	112	112	112	111	112	112	100%	100%	100%	99%	100%
**Coding** **(unit, amino acid length)**											
Gag	500	500	500	504	500	500	100%	100%	100%	101%	100%
Protease	99	99	99	99	99	99	100%	100%	100%	100%	100%
Reverse transcriptase	440	267	440	440	440	440	100%	61%	100%	100%	100%
RNaseH	120	NA	120	120	120	120	100%	NA	100%	100%	100%
Integrase	288	NA	288	288	288	288	100%	NA	100%	100%	100%
Vif	192	192	192	192	192	192	100%	100%	100%	100%	100%
Vpr	96	37	78	96	96	96 *	100%	39%	81%	100%	100%
Vpu	22	22	82	81	82	82	27%	27%	100%	99%	100%
Env	861	859	856	849	854	856	101%	100%	100%	99%	100%
GP120	486	484	481	474	479	481	101%	101%	100%	99%	100%
GP41	345	345	345	345	345	345	100%	100%	100%	100%	100%
Tat	86	86	86	101	86	101 *	85%	85%	85%	100%	85%
Rev	116	100	116	116	116	116	100%	86%	100%	100%	100%
Nef	206	206	123	216	206	206 *	100%	100%	60%	105%	100%
HIVSeqinR verdict	Intact	PrematureStop	Intact	Intact	Intact		Intact	PrematureStop	Intact	Intact	Intact

* These expected values are based on manually removing mutations associated with defects in HXB2; ** LAV was the parent HIV strain for all of ACH-2, 8E5/LAV, and the 3′ end of NL4-3; *** GenBank Accession Number for HXB2 is K03455. Red fonts indicate strain-specific values that are <95% of the expected value settings in HIVSeqinR.

## Data Availability

Not applicable.

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
