# Peer review of "Chemistry and Bioinformatics Considerations in Using Next-Generation Sequencing Technologies to Inferring HIV Proviral DNA Genome-Intactness"

_viruses, 2021, doi:10.3390/v13091874_

Round 1

Reviewer 1 Report

Overall, this is a helpful review that highlights next generation sequencing methodology FLIP-seq and MIP-seq. The focus of the review are techniques so it does read like a methods paper; however, this also highlights some caveats of the approaches. The figures are helpful so that even a more general audience can appreciate the approaches. Some minor comments/suggestions.

Line 75 I think the author meant activation/stimulation rather than "single round infection"

In this same section, there is discussion of the limitations of PCR and ddPCR approaches in detecting defective vs intact proviruses. This neglects the fact of several recent papers that have used multiple probes and multiplexing to more fully characterize proviral status by ddPCR.

A discussion around the challenge of limiting proviral DNA template, especially in patient samples is warranted.

Maybe as part of the conclusion, a more comprehensive discussion of how this technilogy is translated or used to monitor proviral landscapes in patients and re-emphasizing why this is important would be helpful. 

Some proof-reading and minor editing would make the manuscript, especially the introduction, more readable.

Author Response

Overall, this is a helpful review that highlights next generation sequencing methodology FLIP-seq and MIP-seq. The focus of the review are techniques so it does read like a methods paper; however, this also highlights some caveats of the approaches. The figures are helpful so that even a more general audience can appreciate the approaches. Some minor comments/suggestions.

Thank you for your very constructive feedback.  

Line 75 I think the author meant activation/stimulation rather than "single round infection"

Response:  The sentence has been revised to, “but is labor-intensive and has been shown that a single round of activation is insufficient to reactivate all replication-competent genome-intact proviruses.”

In this same section, there is discussion of the limitations of PCR and ddPCR approaches in detecting defective vs intact proviruses. This neglects the fact of several recent papers that have used multiple probes and multiplexing to more fully characterize proviral status by ddPCR.

Response:  This section has been extensively revised to include a discussion on qPCR/ddPCR assays that quantify intact versus defective HIV DNA genomes.

A discussion around the challenge of limiting proviral DNA template, especially in patient samples is warranted.

Response:  Thank you.  I also agree a discussion on sample availability is necessary.  It has been added to the end of section 4 (line 279-285).

Maybe as part of the conclusion, a more comprehensive discussion of how this technology is translated or used to monitor proviral landscapes in patients and re-emphasizing why this is important would be helpful.

Response:  Thank you.  The revised conclusion now includes examples of how these assays have been applied and why they are important.

Some proof-reading and minor editing would make the manuscript, especially the introduction, more readable.

Response:  The article has been revised for English language style and spelling.

Reviewer 2 Report

This commentary gives a short and well-written overview on the novel sequencing technologies that are used to investigate the intactness of the HIV proviral genome. It is a very educational document, of interest for HIV scientists who are not directly involved in the HIV cure research, but who are eager to get acquainted with these new technologies.

Author Response

This commentary gives a short and well-written overview on the novel sequencing technologies that are used to investigate the intactness of the HIV proviral genome. It is a very educational document, of interest for HIV scientists who are not directly involved in the HIV cure research, but who are eager to get acquainted with these new technologies.

Response:  Thank you for your positive feedback.  The article has been revised for English language style and spelling.